# *Leishmania infantum* (JPCM5) Transcriptome, Gene Models and Resources for an Active Curation of Gene Annotations

**DOI:** 10.3390/genes14040866

**Published:** 2023-04-04

**Authors:** Esther Camacho, Sandra González-de la Fuente, Jose Carlos Solana, Laura Tabera, Fernando Carrasco-Ramiro, Begoña Aguado, Jose M. Requena

**Affiliations:** 1Centro de Biología Molecular Severo Ochoa (CSIC-UAM), Departamento de Biología Molecular, Instituto Universitario de Biología Molecular (IUBM), Universidad Autónoma de Madrid, 28049 Madrid, Spain; esther.camacho.cano@gmail.com (E.C.); jcsolana@cbm.csic.es (J.C.S.); 2Centro de Biología Molecular Severo Ochoa (CSIC-UAM), Genomic and NGS Facility (GENGS), 28049 Madrid, Spain; sandra.g@cbm.csic.es (S.G.-d.l.F.); ltabera@cbm.csic.es (L.T.); fcarrasco@cbm.csic.es (F.C.-R.); baguado@cbm.csic.es (B.A.); 3Centro de Investigación Biomédica en Red (CIBERINFEC), Instituto de Salud Carlos III, 28029 Madrid, Spain

**Keywords:** *Leishmania*, transcriptome, gene expression, gene models, Mendeley data, Wikidata

## Abstract

*Leishmania infantum* is one of the causative agents of visceral leishmaniases, the most severe form of leishmaniasis. An improved assembly for the *L. infantum* genome was published five years ago, yet delineation of its transcriptome remained to be accomplished. In this work, the transcriptome annotation was attained by a combination of both short and long RNA-seq reads. The good agreement between the results derived from both methodologies confirmed that transcript assembly based on Illumina RNA-seq and further delimitation according to the positions of spliced leader (SAS) and poly-A (PAS) addition sites is an adequate strategy to annotate the transcriptomes of *Leishmania*, a procedure previously used for transcriptome annotation in other *Leishmania* species and related trypanosomatids. These analyses also confirmed that the *Leishmania* transcripts boundaries are relatively slippery, showing extensive heterogeneity at the 5′- and 3′-ends. However, the use of RNA-seq reads derived from the PacBio technology (referred to as Iso-Seq) allowed the authors to uncover some complex transcription patterns occurring at particular loci that would be unnoticed by the use of short RNA-seq reads alone. Thus, Iso-Seq analysis provided evidence that transcript processing at particular loci would be more dynamic than expected. Another noticeable finding was the observation of a case of allelic heterozygosity based on the existence of chimeric Iso-Seq reads that might be generated by an event of intrachromosomal recombination. In addition, we are providing the *L. infantum* gene models, including both UTRs and CDS regions, that would be helpful for undertaking whole-genome expression studies. Moreover, we have built the foundations of a communal database for the active curation of both gene/transcript models and functional annotations for genes and proteins.

## 1. Introduction

The term leishmaniasis refers to a group of neglected tropical diseases caused by protozoan parasites of the genus *Leishmania*, which belongs to the order Trypanosomatida. These diseases may have different clinical manifestations, the main ones being cutaneous leishmaniasis (CL), mucocutaneous leishmaniasis (MCL), and visceral leishmaniasis (VL). Globally, leishmaniasis represents a serious public health concern; according to WHO estimations, around 1 million new cases occur annually worldwide [1]. CL courses with skin ulcers, nodular lymphangitis, and/or satellite lesions, and it is the most common form of leishmaniasis; MCL causes skin and mucosal ulcers, and VL, which is the most severe clinical form, occurs with sporadic fever, weight loss, anemia and enlargement of the liver and spleen [2]. VL is due to the viscerotropic species *Leishmania infantum* and *Leishmania donovani,* and the disease results in fatality if untreated.

In 2007, the first draft of the *L. infantum* genome was published [3], and it served for conducting genome-wide gene expression analysis [4], proteomic studies [5], and mining the annotated proteome for predicting MHC-restricted epitopes [6]. Nevertheless, although a valuable resource, the genome assembly was not continuous, and most of the chromosomes contained gaps and indeterminate sections in their sequence. In 2017, an improved genome assembly was generated for the same strain (JPCM5) of *L. infantum* by a combination of second and third-generation sequencing methodologies, resulting in a continuous sequence for the 36 chromosome-sized contigs that constitute the genome of this species [7]. Currently, this new assembly has been incorporated as the reference genome for this species in TriTrypDB [8], and its predicted proteome is available in the UniProt database [9].

Regulation of gene expression in *Leishmania* and related trypanosomatids, such as *Trypanosoma brucei* and *Trypanosoma cruzi*, operates utmost exclusively at the post-transcriptional level [10,11,12]. To accommodate such a peculiar strategy of regulation, genes are grouped in a strand-specific orientation and transcribed in large polycistronic transcripts [13]. These transcripts are co-transcriptionally processed into individual mature RNAs by the concerted action of two molecular processes: trans-splicing and polyadenylation [14]. The trans-splicing mechanism, which shares similarities with cis-splicing, leads to the addition of a common 39 nucleotide sequence (known as a spliced leader or mini-exon) at the 5′ of every protein-coding transcript [15]. In contrast, cis-splicing events are very rare in trypanosomatids, and only two cases have been reported to date [16,17,18].

Given that transcription at the per-gene level does not exist in trypanosomatids, the current view is that regulation of gene expression relies on mRNA stability and translation efficacy, two processes modulated by trans-acting factors that bind to specific cis-acting elements on particular sets of target mRNAs [19]. These cis-regulatory elements are frequently located in mRNA untranslated regions (UTRs). In this context, the annotation of complete gene models rather than coding sequences alone is paramount to address questions related to mechanisms of gene expression regulation in trypanosomatids [20].

Transcriptomic studies also contribute to improving gene annotations in two additional manners. On the one hand, the precise mapping of the 5′-ends of transcripts may provide information on errors in open reading frame (ORF) assignments if the transcript start point is located downstream of the originally assigned initiator ATG. On the other hand, transcriptomics approaches serve to identify previously non-annotated genes, mainly those having non-coding functions [21].

In previous works, our group determined the poly-A+ transcriptome for the species *Leishmania major* [21] and *L. donovani* [18]. The third species having its transcriptome delineated was *L. mexicana* [22]. Here, we are contributing the poly-A+ transcriptome for the species *L. infantum* and the annotation of the corresponding gene models. Moreover, we have determined relative expression levels for all the transcripts, and identified particular transcripts whose steady-state levels are markedly different in the *L. infantum* and *L. major* species.

## 2. Materials and Methods

### 2.1. Leishmania Parasites

The *L. infantum* strain MCAN/ES/98/LLM-724 was isolated by Dr. J. Moreno’s group (WHO Collaborating Centre for Leishmaniasis, Centro Nacional de Microbiología, Instituto de Salud Carlos III, Madrid, Spain) from a dog suffering from visceral leishmaniasis [23]. A clone (named JPCM5) from this strain was chosen as the *L. infantum* reference strain (TriTrypDB) and widely distributed among laboratories. The promastigote form was cultured at 26 °C in M199 medium (Sigma-Aldrich, St. Louis, MO, USA) supplemented with 40 mM HEPES (pH 7.4), 100 μM adenine, 10 µg/mL hemin, 0.0001% biotin, 0.2 ng/mL biopterina, and 10% heat-inactivated fetal calf serum. Cryopreserved aliquots were used and kept in culture for a maximum of 20–25 passages.

### 2.2. RNA Isolation

RNA for Illumina sequencing was prepared from around 4 × 10^8^ promastigotes in the late logarithmic phase. At the time of harvesting, the cell densities of the three biological replicates were 9 million/mL (sample Jpc1), 6.8 million/mL (Jpc2), and 7 million/mL (Jpc3). After harvesting by centrifugation, the pellet was suspended in 1 mL of TRI Reagent (Sigma-Aldrich, product No. T9424). The manufacturer’s instructions were followed. Samples were kept at −70 °C for a week before proceeding with the phase separation. After thawing, 0.2 mL of chloroform was added, and the mixtures were shaken vigorously for 15 s. After centrifugation, the colorless upper aqueous phase (containing RNA) was processed to obtain the RNA.

In order to extract the RNA for Iso-Seq sequencing, the NucleoSpin kit (Cultek S.L.U, Madrid, Spain) was used. Around 5 × 10^7^ logarithmic phase promastigotes were used.

RNA samples were suspended in DEPC-treated water, and their concentrations were determined using the Nanodrop ND-1000 (Thermo Fisher Scientific, Rockford, IL, USA); the samples showed A260/A280 ratios higher than 2.0. In addition, RNA integrity was checked in a bioanalyzer (Agilent 2100) before proceeding with cDNA synthesis.

### 2.3. Illumina RNA-Seq and Data Processing 

Library construction from poly-A+ RNA and paired-end library sequencing was performed at the Centro Nacional de Análisis Genómico (CNAG-CRG, Spain) using Illumina HiSeq 2000 (v4) technology. As a result, we received 99,117,195 (2 × 76 nt) stranded reads.

The transcriptome of *L. infantum* (JPCM5) was annotated following the pipeline described by Rastrojo et al. [21]. As a reference, the 2017 *L. infantum* (JPCM5) genome assembly [7] was used. This version is available in different databases (EBI, NCBI, and TriTryDB), but it also may be downloaded at: http://dx.doi.org/10.17632/rb34cg9xk7.1 (accessed on 31 March 2023). To sum up the pipeline, firstly, a standard quality filtering process of the reads was undergone using FASTQC (www.bioinformatics.babraham.ac.uk/projects/fastqc/ accessed on 27 May 2022). The reads were then mapped against the *L. infantum* genome using Bowtie2 (v2.34.3) with the following parameters: --np 0 --n-ceil L,0,0.02 --rdg 0,6 --rfg 0,6 --mp 6,2 and --score-min L,0,-0.24. Additionally, a search among the unaligned reads was performed using an in-house Python script to locate reads that contained eight or more nucleotides identical to the SL sequence (AACTAACGCT ATATAAGTAT CAGTTTCTGT ACTTTATTG). Then, the SL-derived nucleotides were trimmed from these reads, and the remaining sequences were mapped back to the *L. infantum* (JPCM5). Thus, the coordinates for the SL-addition sites (SAS) were determined. A similar procedure was undergone to identify the polyadenylation addition sites (PAS). For the search of PAS, the presence of an A-string of 5 nucleotides or longer was looked for at the end of the reads. Finally, the transcripts assembled by Cufflinks (v2.2.1) [24] were trimmed according to the positions of the SAS and PAS. Finally, a manual revision and curation of the annotated transcripts was performed using the Integrative Genomics Viewer (IGV, v2.14.0; [25]).

For transcript nomenclature, if the transcript contained a previously annotated gene in the genome, the gene code was maintained, and a T was added at the end of the gene ID (e.g., transcript LINF_010005000-T refers to the transcript derived from gene LINF_010005000). When no previously annotated gene existed, the transcripts were labeled with an intercalated serial number between those of their flanking transcripts with associated CDSs. When several transcripts were derived from a given gene, a serial number was added (-T1, -T2, and so on).

### 2.4. Iso-Seq Analyses

First-strand synthesis of cDNA was prepared from 0.2 µg of *L. infantum* (JPCM5) total RNA using the SMARTer cDNA synthesis kit (Clontech, Göteborg, Sweden), which incorporates a poly-T oligonucleotide to prime the DNA synthesis from the 3′-end of mRNAs. One-tenth of the cDNA synthesis reaction volume was used for PCR cycle optimization using as the sole primer the SMARTer II A oligonucleotide (Clontech). For PCR amplification, the Advantage 2 PCR kit (Clontech) and PrimeSTAR GXL DNA polymerase (Clontech) were used. After visualization of the PCR products on a 1% agarose gel, it was considered that optimal amplification was attained after 26 cycles of PCR. Then, a large-scale PCR reaction was prepared, and 13.67 µg of double-strand cDNA were obtained; afterward, this material was fractionated using AMPure XP beads (Beckman Coulter, Palo Alto, CA, USA), and the 0.4 × AMPure fraction was chosen for sequencing.

The library was prepared using Pacific Biosciences (PacBio) protocol for Iso-Seq™ Template preparation for Sequel Systems. Then, the library was sequenced on PacBio Sequel instrument using Sequel Polymerase v3.0, SMRT cells v3 LR, and Sequencing chemistry v3.0. Loading was conducted by diffusion, and sequencing was performed on one SMRT cell. Both services (library preparation and sequencing) were provided by the Norwegian Sequencing Centre (www.sequencing.uio.no; accessed on 3 February 2021), a national technology platform hosted by the University of Oslo.

Iso-Seq analysis was performed using the Iso-Seq pipeline on SMRT Link (v7.0.0.63985, SMRT Link Analysis Services, and GUI v7.0.0.63989) using default settings. Then, the high-quality isoforms obtained were aligned to the *L. infantum* (JPCM5) genome [7] using the GMAP software [26] and processed with the Cupcake ToFu pipeline (sam_to_gff3.py) [https://github.com/Magdoll/cDNA_Cupcake/wiki/; accessed on 15 January 2023] to generate a GFF3 file. Finally, full or incomplete cDNA sequences were separated using an in-house Python script designed to locate the molecules containing the Spliced Leader sequence at the beginning (when they are complete cDNAs).

### 2.5. PCR Assays

For experimental testing of allelic deletion (hemizygosity) affecting genes *LINF_320009700*, *LINF_320009800*, and *LINF_320009900*, the following oligonucleotides were designed: Li329700-Fw (F; #813), 5′-GAACCAGGAG CAGCTTCAAC-3′; Li329700-Rv (R; #814), 5′-AGGTCTGCAG CTCAGGTCAT; Li329900-Rv (R′; #815), 5′-AACCGCTAGA GGCACCAGTA. Phusion Hot Start II DNA Polymerase (Thermo Fisher Scientific, Rockford, IL, USA) was used for amplifications following the supplier’s procedure. *L. infantum* (JPCM5) total DNA (50 ng) was used as template, and PCR was carried out in the presence of 3% DMSO. The PCR profile was: initial denaturation (98 °C for 4 min), 35 cycles consisting of 10 s at 98 °C (denaturation), 30 s at 64.2 °C (annealing), and 1 min at 72 °C (polymerization). A final incubation at 72 °C for 10 min was included. Amplification products were analyzed by electrophoresis on 1% agarose gels in Tris-acetate EDTA buffer. *Hin*dIII-digested phage Ø29 DNA was used as a molecular weight marker [27].

### 2.6. Gene-Model Annotation

The genome sequence, the CDS location, and the transcriptome coordinates were merged to create gene models, which were named according to the transcript ID (see above), but with the “T” eliminated from the transcript ID.

### 2.7. Determination of RNA Levels and Differential Expression from RNA-Seq Data

Counts of RNA-seq reads per transcript were calculated using HTSeq v1.0 with the parameters: intersection-strict and minaqual 1 [28]. Analysis of relative RNA abundance was carried out by the TPM (transcripts per million) method [29].

To compare differential transcript levels between the species *L. infantum* and *L. major*, pairs of orthologous genes were generated using BLAST alignments [29]. A total of 8871 pairs were established, and TPM counts were assigned to each one. Afterward, differential expression levels between orthologous pairs were determined using the SCBN software [30], a tool specifically designed to determine differentially expressed genes (DEGs) between two different species. Those pairs of orthologs having a ratio of TPM counts between 0.995 and 1.005 were used as reference (control) groups.

### 2.8. Data Availability

Transcriptomic raw data have been deposited at the European Nucleotide Archive (ENA; http://www.ebi.ac.uk/ena/ accessed on 21 February 2023). The Illumina RNA-seq data, the transcriptome sequences, together with annotations files were uploaded under the Study accession number PRJEB46649 (ERP130857). Iso-seq raw data and sequences were deposited at ENA under the Study accession number: PRJEB38965.

## 3. Results and Discussion

### 3.1. Delineation of the Poly-A+ Transcriptome for the L. infantum (JPCM5 Strain) Promastigote Stage Based on Illumina RNA-Seq Methodology

For this purpose, total RNA from three biological replicas of logarithmically growing promastigotes was isolated. After poly-A+ selection and Illumina library preparation, stranded mRNA-seq (2 × 75 bp) data were generated in the Illumina HiSeq 2000 platform (see Methods section for further details). After quality filtering, a total of 99,117,195 paired reads were selected for subsequent analyses.

Transcript annotation was conducted following the pipeline described elsewhere [21] and briefly summarized in the Methods section. For researchers interested in conducting similar studies but lacking the needed bioinformatics skills, we recommend the use of the SLaP mapper web server developed by Fiebig and co-workers [31]. This server, starting with raw read data from paired-end Illumina RNA-seq (for any trypanosomatid), performs read processing, mapping, quality control, quantification, and other analyses in a fully automated pipeline. Annotation was based on the *L. infantum* (JPCM5 strain) reference genome [7]. As a result, 9646 transcripts were annotated. The presence of SL-nucleotides at the 5′-end of the transcripts allowed the authors to precisely define the 5′-end for 9527 transcripts (98.8%). Moreover, secondary SL-addition sites (SASs) were also identified for 9246 transcripts. However, the presence of a poly-A+ tail could be determined for 819 (8.5%) of the transcripts. For no well-defined technical reasons, the poly-A tail-containing RNA-seq reads are under-represented in the Illumina libraries, as noted by us and others in previous studies [18,21,32]. Fortunately, this limitation was not observed using the Iso-Seq methodology (see below); thus, summing up the results from the Iso-Seq data, poly-A+ addition sites (PASs) could be defined for 57.37% of the transcripts. In addition, the existence of alternative PASs for many transcripts was found. The complete transcriptome, including the positions of SASs and PASs, is provided in Appendix A.

Among the 9646 annotated transcripts in *L. infantum*, 8509 contained predicted CDS, and the rest may be considered as putative non-protein coding transcripts. Similar numbers of coding and non-coding transcripts have been annotated in other *Leishmania* species [18,22,33].

### 3.2. Refinement of the Transcriptome by the Iso-Seq Methodology

To check the accuracy of the transcriptome determined by RNA sequencing using Illumina short-reads, we generated RNA-seq data based on the use of PacBio long RNA-seq reads (Iso-Seq methodology). In all, 19,085 distinct molecules were sequenced, and 1577 of them represented complete transcripts (SL sequence was found at their 5′-end and poly-A at their 3′-end). As a whole, we identified Iso-Seq transcripts derived from 5967 different genes. The coincidence between the SASs determined by the Illumina RNA-seq and those found in the complete molecules sequenced by Iso-Seq was found in 95.7% of the transcripts. Only 56 Iso-Seq transcripts contained a SAS not previously mapped in the Illumina RNA-seq reads. In sum, these results allowed us to conclude that transcript annotation by Illumina RNA-seq is a very accurate approach for transcriptome delineation in intron-less organisms such as *Leishmania*. In addition, as cDNA synthesis is primed by a poly-T-containing oligonucleotide, the Iso-Seq methodology was found useful in improving the PAS mapping of transcripts.

In addition, the Iso-Seq data served to uncover a few previously unnoticed peculiarities of the transcription pattern at some *L. infantum* loci. Thus, on the one hand, we found evidence that the processing of polycistronic transcripts may not occur in a strictly sequential manner in some loci. Figure 1A shows an example in which two consecutive genes, *LINF_130006600* and *LINF_130006700*, would be included in a bicistronic transcript based on the existence of Iso-Seq reads expanding both genes. According to the pipeline used for transcript annotation, based on the mapping of SAS, two independent transcripts (LINF_130006600-T and LINF_130006700-T) could be annotated. However, the analysis of Iso-Seq molecules showed that a bicistronic transcript (LINF_130006700-600-T) also exists. In fact, as a PAS for transcript LINF_13T0006700-T could not be mapped, it cannot be excluded that this gene is expressed only by the bicistronic transcript. The existence of other bicistronic transcripts was also evidenced for the following pairs of consecutive genes: *LINF_120010600* and *LINF_120010700*; *LINF_120012350* and *LINF_120012400*; *LINF_230015800* and *LINF_230015900*; *LINF_250008200* and *LINF_250008300*; *LINF_250012600* and *LINF_250012700*; *LINF_260012600* and *LINF_260012700*; *LINF_260012900* and *LINF_260013000*; *LINF_270019200* and *LINF_270019300*.

Another remarkable finding is depicted in Figure 1B, which shows a case in which three different transcripts were generated from the gene *LINF_240027300*. In our pipeline of transcript annotation based on Illumina RNA-seq reads, when SASs were found within the 3′-UTR, they were considered possible spurious events and ignored for transcript delineation. Thus, in this case, only a transcript (LINF_240027300-T), expanding the complete gene, was annotated. However, the analysis of Iso-Seq reads showed that a mature transcript, having an SL sequence at its 5′ and a poly-A tail at the 3′-end, is being generated within the 3′-UTR of the *LINF_240027300* gene. Moreover, a second transcript, expanding the upstream region of the gene, was also found, together with a third transcript, which expands the complete gene region. Therefore, from these data, we concluded that three mature transcripts are generated from the *LINF_240027300* gene: LINF_240027300-T1, LINF_240027300-T2, and LINF_240027300-T3 (Figure 1B). Other genes having mature transcripts derived from the 3′-UTR are *LINF_120007000*, *LINF_130020800*, *LINF_190015900*, *LINF_24T0027300*, *LINF_270007800*, *LINF_280029400*, and *LINF_360030000*.

### 3.3. Identification of an Allelic Deletion (Hemizygosity) Affecting Genes LINF_320009700, LINF_320009800 and LINF_320009900

Another remarkable finding was observed when analyzing in detail the Iso-Seq transcripts mapping on the genomic region in which genes *LINF_320009700*, *LINF_320009800*, and *LINF_320009900* are located (Figure 2). Two sets of transcripts were found. On the one hand, five transcripts (truncated at its 5′-end) were identified as derived from gene *LINF_320009800* (coding for a Leucine Rich Repeat (LRR)-containing protein). On the other hand, the sequence of the other three transcripts evidenced their chimeric nature: they contained the coding region of gene *LINF_320009900* but ended within the 3′-UTR of gene *LINF_320009700* (Figure 2A). In fact, one of those Iso-Seq molecules was complete as it contains SL-derived sequences at its 5′-end. A plausible explanation was that an intrachromosomal recombination event might be occurring in one of the chromosomes, leading to the excision of the region expanding genes *LINF_320009700* (coding for prostaglandin f synthase) and *LINF_320009800*. This hypothesis was supported by the decrease in the DNA-seq coverage observed after the mapping of PacBio reads (Figure 2B). Interestingly, two 695 nt direct repeats were found flanking the genomic region with lower DNA-seq coverage. To find another experimental support, specific oligonucleotides were designed to enable PCR to amplify the region in the study (Figure 2C). The presence of a PCR amplification band of 1601 nt in length with the oligonucleotides located at both ends of the genomic region (oligonucleotides F and R’; panel C1) demonstrated that the hypothesized intrachromosomal deletion was a real event. Moreover, a faint band of about 8.5 kb (8692 nt according to the sequence) was also visible in the gel after longer exposure (panel C2, arrow), indicating that both alleles (complete and deleted) co-exist. Additionally, a re-analysis of the PacBio DNA-seq reads showed that around half of the reads expanding this genomic region would derive from the chromosome having the deleted allele. In summary, the experimental data suggest that promastigotes experience an intrachromosomal reorganization leading to the creation of a chimeric transcript for gene *LINF_320009900* (coding for a putative replication termination factor) and simultaneously lowering (half) the gene doses for *LINF_320009800* and *LINF_320009700*. Further experiments are warranted to ascertain the functional purpose and consequences of this allelic heterozygosity.

### 3.4. Setting Up Gene Models

During the last decade, many genomes of trypanosomatids have been sequenced and annotated (see TriTrypDB, a functional genomic resource specialized in trypanosomatids [8]), but annotation of complete genes, including their boundaries, is almost absent in popular databases. For instance, in TriTrypDB, this information is only available for *T. brucei* and, more recently, for *L. major*; for the rest of trypanosomatids, annotated gene sequences are restricted to the coding sequences (CDS), and consequently genes and CDS are often used as synonymous terms. However, genes contain UTRs, which sometimes are longer than the CDS and that are involved in regulating the fate of the gene-encoded mRNA molecules [34,35]. In previous works, our group generated gene models for *L. donovani* [18] and *L. major* [33]; the information may be visualized and downloaded from the Leish-ESP web page (http://leish-esp.cbm.uam.es accessed on 31 March 2023) and also in TriTrypDB repository for the *L. major* genome.

Here, we have merged the *L. infantum* poly-A+ transcriptome generated in this study with the de novo assembled genome reported previously [7] to generate the gene models for the *L. infantum* JPCM5 strain, which is the reference strain for this species (TryTrypDB). Figure 3 shows a scheme illustrating the steps followed to create the gene models. Firstly, after mapping RNA-seq reads to the reference genome, transcripts were assembled by the Cufflinks tool [24] and trimmed according to the position of the main SAS and PAS. Afterward, the automatically annotated CDS (using the Companion web server [36]) was located in the corresponding transcript. Most of the CDS fitted well in the transcripts, but sometimes, as illustrated in Figure 3D, the annotated CDS started upstream of the transcript. In these cases, the CDS sequence was manually inspected to find the first in-frame ATG initiation coding within the transcript, and consequently, the CDS was shortened (Figure 3E). Finally, CDS and transcript sequences were merged to create the gene model consisting of 5′-UTR, CDS, and 3′-UTR moieties (Figure 3F). The gene models will be deposited in the TriTrypDB database and are currently available as Mendeley datasets (see below).

### 3.5. Relative Expression Levels of Transcripts in L. infantum Promastigotes

Once accurate gene models were available, we analyzed the expression levels for each one of the annotated transcripts by using RNA-seq data derived from three biological samples consisting of *L. infantum* promastigotes (see Materials and Methods for additional details). The use of replicates allowed the authors to estimate mean values and standard deviations; also, these expression levels may be used to make comparisons with the expression levels of particular transcripts in promastigotes from other *Leishmania* species (see below). Relative expression levels were determined for 8476 protein-coding transcripts (see Appendix A). Table 1 shows the top 40 transcripts according to their relative abundance. Although high transcript levels are not a demonstration of high levels of the encoded proteins, the finding that most expressed transcripts encode for abundant structural proteins (histones, tubulins, and ribosomal proteins) suggested a likely correlation between transcript levels and protein abundance in many cases.

Among the most expressed transcripts, apart from those coding for well-known structural proteins, there are a few other transcripts whose encoded proteins merit being briefly commented on. The fourth most abundant transcript (LINF_250014900-T) codes for a cyclophilin A (named Cyp19) that exhibits peptidyl-prolyl cis/trans isomerase activity as reported elsewhere [37]. Peptidyl-prolyl isomerization is a rate-limiting step in protein folding, a feature that might explain the abundance of this transcript (and perhaps the encoded protein). Transcript LINF_270006200-T would be encoding for the ortholog to *T. brucei* ZFP3 protein, which has been associated with the translational apparatus and is involved in trypanosome life-cycle development [38]. Transcript LINF_130009400-T encodes for Alba1 protein, an RNA-binding protein postulated to regulate mRNA translation by interacting with ribosomal subunits and translation factors [39]. An inosine-guanosine transporter is encoded by the 15th most abundant transcript (LINF_360026000-T), and this may be related to the parasite dependence on imported purine nucleobases as *Leishmania* is incapable of de novo synthesis of the purine ring [40]. Transcript LINF_230018500-T encodes a protein of unknown function but conserved in sequence among the different *Leishmania* species. In *Leishmania*, due to the evolutionary distance regarding the eukaryotic model organisms [41], a large fraction of the proteome belongs to the category of proteins of unknown function because of the lack of sequence similarity to proteins from other groups of eukaryotes. Kinetoplastid membrane protein-11 (KMP11), with a copy number of around 1 million molecules per cell [42], is encoded by another abundant transcript (LINF_350027400-T). The 37th most abundant transcript is LINF_230005400-T which encodes for the peroxidoxin mTXNPx, which also acts as a mitochondrial chaperone involved in preventing protein aggregation [43]. Finally, a transcript derived from the HSP70-type II gene was ranked 39th among the most abundant mRNAs in *L. infantum* promastigotes. Noteworthy, this abundant transcript is known to be non-translated at normal growth temperatures, but it is stored at ambient temperature (when the parasite is inside the insect host) to be translated during heat shock [44]. Thus, this is an example in which a high transcript level does not mean a high level of protein translation. Another remarkable finding derived from this list of most abundant transcripts is the presence of different transcripts coding for the same protein (e.g., histones H1, H4, H2B, H3, H2A, and α tubulin), but their abundance is significantly distinct. This fact provides evidence that transcript expression levels for genes coding for the same protein may be different; however, it should be remarked that those differences could not be appreciated if RNA-seq reads were mapped solely against the gene coding regions.

### 3.6. Species-Specific Differences in the Steady-State Transcript Abundance

Different species of *Leishmania* are largely responsible for the diverse clinical manifestations of leishmaniasis [45]. For instance, the species *L. donovani* and *L. infantum* mostly cause visceral leishmaniasis (often deadly if untreated), whereas *L. major* infection leads to self-healing cutaneous leishmaniasis. Nevertheless, these species are morphologically indistinguishable, their genomes are largely syntenic, and few are species-specific genes [46]. For instance, only fifteen out of the 8405 protein-coding genes annotated in the *L. donovani* genome could not be identified in the *L. major* genome [18]. Thus, it is possible that differential gene expression levels may contribute to these differences in the virulence developed by the different *Leishmania* species.

In a previous study, we analyzed the transcriptome for *L. major* promastigotes [33], following an equivalent experimental design to that used here to analyze the *L. infantum* transcriptome. In both studies, we prepared RNA-seq data from three biological replicates consisting of axenic promastigotes growing at the mid-logarithmic phase. When transcripts were ranked by abundance, there was a generally high agreement between those determined in *L. infantum* (Table 1) and those most abundant in *L. major* promastigotes [33]. However, when a detailed analysis of transcript-by-transcript was conducted, a few orthologous transcripts showed steady-state levels markedly different among these two *Leishmania* species. Table 2 lists those ortholog transcripts having differences of four or more times in the steady-stated between both species. Apart from transcripts coding for proteins of unknown function/hypothetical, it is remarkable the presence of several transcripts coding for membrane proteins and RNA binding proteins. Nevertheless, none of these RNA-binding proteins have been studied to date in *Leishmania*. Functional characterization of the ortholog to RBP10 (*LINF_230015000*) in *T. brucei* has been reported; in this *Leishmania*-related parasite, this protein was found to be required for the differentiation from the procyclic stage (insect forms) to the bloodstream forms [47]. Another differentially expressed transcript, LINF_120010100-T, which is about 5-fold more abundant in *L. infantum* than in *L. major*, was studied by the Clos’ laboratory some time ago [48]. The encoded protein was dubbed LdGF1 (growth factor 1) because its overexpression allowed the parasites to recover faster from the stationary phase. In contrast, but remarkably, these authors found that overexpression of LdGF1 in *L. major* resulted in reduced virulence when inoculated in BALB/c mice. From these findings, it may be postulated a species-specific role for this protein is in agreement with the different expression levels described here. Nevertheless, a degree of caution should be exercised as transcript levels and protein levels do not always correlate in *Leishmania*.

### 3.7. Active Curation of Gene Models and Annotations Based on Wikidata and Mendeley Data Platforms

The L. infantum (JPCM5 strain) genome assembled by our group [7] is currently the reference genome for this species in TriTrypDB, a section of the VEuPathDB repository [49] containing invaluable resources and tools for studying genomes of trypanosomatids.

Although we plan to deposit into TriTrypDB the gene models established in this work, the process is not a straightforward task. Additionally, it may not be adequate for general repositories (ENA, NCBI) to update a genome sequence every time the annotation of a sole gene is changed. However, genomic annotations are being continuously improved on the basis of new experimental data, such as proteomics, for instance [50]. In addition, many groups working on particular genes/proteins are contributing to greatly increasing our knowledge about them, and, in consequence, functional annotations of genes/proteins are continuously improved, and these should be registered in the databases as soon as possible and the contributors recognized. To fill this gap, we have created structured data for L. infantum gene models exploiting the special features of Wikidata (https://www.wikidata.org/ accessed on 31 March 2023) and Mendeley data (https://data.mendeley.com/ accessed on 31 March 2023) repositories. Both are free and secure cloud-based repositories aimed to store data that are immediately accessible to everyone. Users can explore both repositories, which are interlinked, simply by browsing the database by gene IDs and/or functional annotations; moreover, for the Wikidata entries, users can contribute with annotations using the simple and intuitive interface that this repository provides. The Mendeley data entries also contain the sequences for genes, CDS, and proteins. The entries in both repositories also inform on articles in which a particular gene/protein has been studied. Wikidata entries also include links to information available at the TriTryDB and UniProt repositories. In summary, the rapid accumulation of new data and knowledge needs to be quickly incorporated to disseminate the improvements in gene and protein annotations, and these tasks may be accomplished in a communal manner by the use of these repositories. However, these repositories are not substitutive of either general or dedicated repositories (i.e., TriTrypDB) that contain bioinformatics tools of enormous value for research activities. The goal is that Wikidata/Mendeley data entries serve for quick updating of gene/protein annotations, but researchers should be aware that in order to preserve the information in a secure manner, it must be incorporated finally into the general repositories.

## 4. Conclusions

We used second and third-generation NGS methodologies to sequence poly-A+ RNA-Seq, and the results allowed the authors to generate an L. infantum (JPCM5) transcriptome at single-nucleotide resolution. As a result, 9646 transcripts were annotated, most having protein-coding regions (CDS); however, CDS could not be annotated for 1175, even though they are SL-spliced and polyadenylated. In consequence, these were categorized as non-coding RNAs (ncRNAs).

The use of long RNA-seq reads (Iso-Seq; PacBio), apart from confirming the accuracy of the transcriptome delineated by the use of short RNA-seq reads (Illumina), allowed to show peculiar transcription patterns occurring at some loci. As a result, we have identified the existence of stable bicistronic transcripts and the production of fully processed transcripts derived from 3′-UTRs of some genes. Thus, the complexity of the Leishmania transcriptome may be higher than expected.

To address transcriptomics changes at either a genomic scale or a single gene, the availability of accurate gene models is paramount. For this purpose, CDS and transcripts annotations were merged to produce the L. infantum gene models, in which 5′- and 3′-UTRs were defined. In order to speed up the availability of this information to interested people, we have created individual gene entries at Wikidata and Mendeley data repositories, where they can be inspected and downloaded and may be curated in a communal manner.

Finally, we have conducted a comparative analysis searching for transcripts having a differential expression in the species L. major (causing cutaneous manifestations) and L. infantum (causing visceral affectation). Although the gene content is almost identical and gene synteny among both genomes is absolute, we have identified a set of transcripts having marked differences in abundance. These differences in expression might be important in terms of different pathogenicity between both species.

## Figures and Tables

**Figure 1 genes-14-00866-f001:**
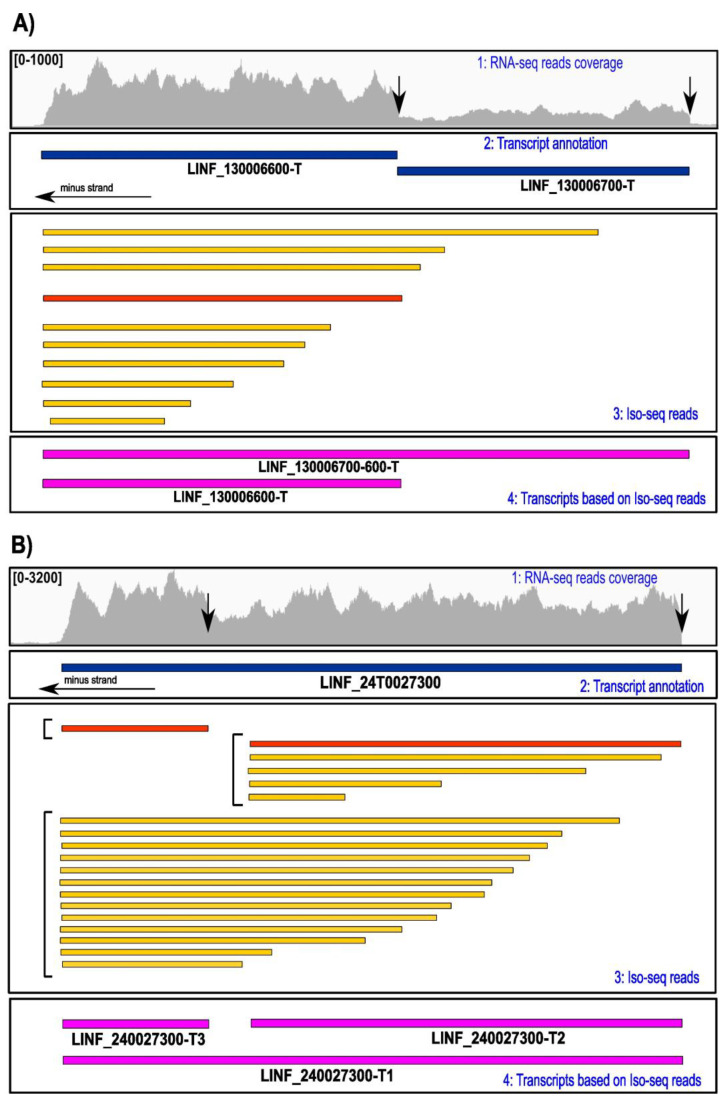
Complex transcription patterns were observed at particular *L. infantum* loci. (**A**) According to Illumina RNA-seq reads coverage and position of SL-addition sites (black vertical arrows), a single transcript was deduced to be expressed from genes *LINF_130006600* and *LINF_130006600* (horizontal arrow shows the transcriptional direction). The existence of a complete Iso-Seq transcript (molecule shaded in red) derived from gene *LINF_130006600* indicated that this gene is expressed into a mature individual RNA. However, no evidence of an individual mRNA for gene *LINF_130006700* was found, suggesting that this gene may be expressed as a bicistronic transcript expanding both genes (transcript LINF_130006700-600-T). (**B**) Illumina RNA-seq reads coverage on gene *LINF_240027300* suggested the existence of a sole mRNA molecule for this gene. However, the presence of two complete Iso-seq molecules indicated that two stable mRNAs are produced (LINF_240027300-T2 and LINF_240027300-T3) together with a third molecule expanding the complete gene (LINF_240027300-T1).

**Figure 2 genes-14-00866-f002:**
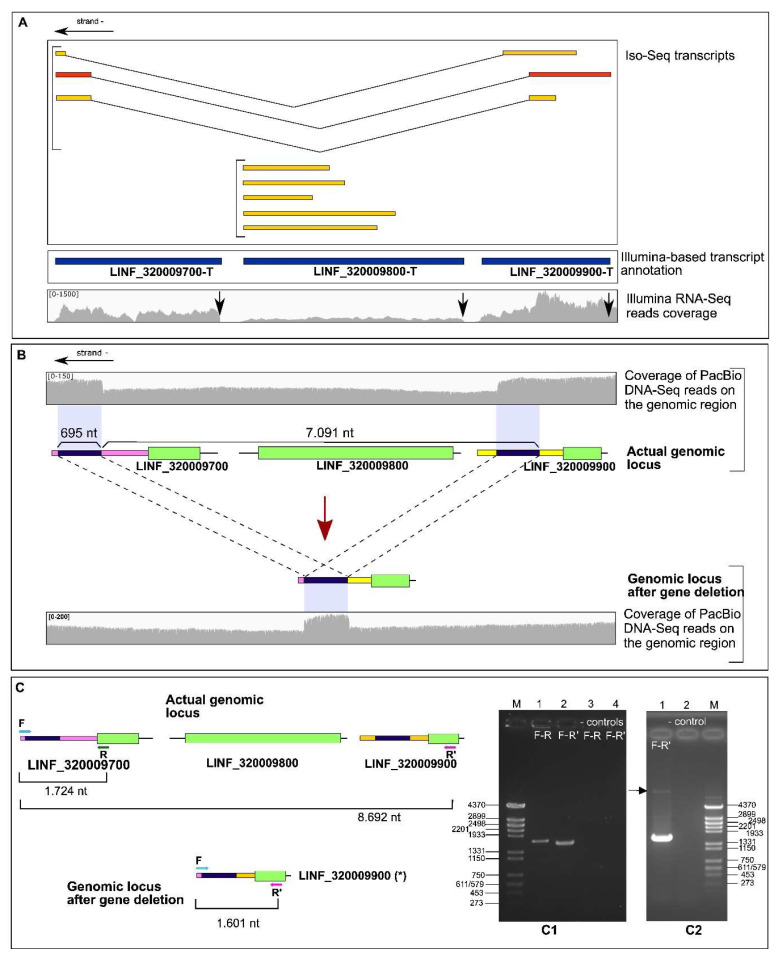
Allelic heterogeneity on a genomic region of *L. infantum* chromosome 32. (**A**) Two types of transcripts (Iso-Seq molecules) were identified: one derived from gene *LINF_320009800* and the other showing a chimeric nature (derived from separated genomic regions). (**B**) The existence of a lower number of PacBio DNA-seq reads in the region-expanding genes *LINF_320009700*, and *LINF_320009800* suggested an intrachromosomal deletion event affecting one allele. The hypothesized structure for both alleles is depicted together with the PacBio DNA-seq reads coverage on both regions. The existence of a direct repeat of 695 nt in length is denoted by black boxes. (**C**) Experimental testing by PCR of the deletion event. Two pairs of oligonucleotides (F-R and F-R’) were used to amplify the 3′-UTR of gene *LINF_320009700* and the whole genomic region (genes *LINF_320009700*, *LINF_320009800* and *LINF_320009900*), respectively. The PCR products were separated on a 1% agarose gel (panel C1). A band of around 8.7 kb was also observed after PCR amplification with oligonucleotides F and R’ (panel C2). At the boundaries of gels are shown the sizes (in nucleotides) of the *Hin*dIII-digested Ø29 phage DNA bands used as a molecular marker (M). Lanes 3 (gel C1) and 2 (gel C2) are controls in which PCR was performed in the absence of a DNA template.

**Figure 3 genes-14-00866-f003:**
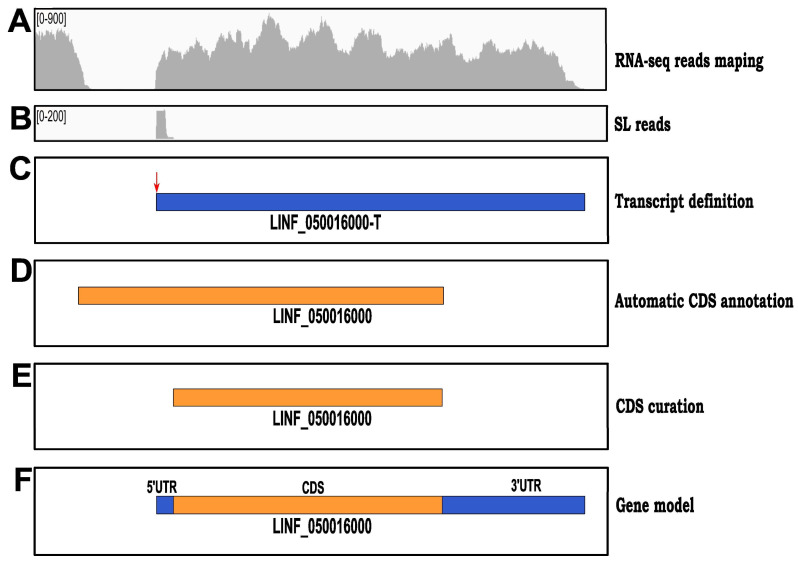
Steps for creating gene models. (**A**) Transcripts are assembled from Illumina RNA-seq reads. (**B**) SL-containing RNA-seq reads are used to map the SL-addition sites (SASs); the main SAS (vertical arrow) is assigned to that supported by the largest number of SL-containing RNA-seq reads, and the rest as categorized as alternatives (see Appendix A for complete lists of main and alternative SASs). (**C**) Transcript is trimmed according to the position of the main SAS. (**D**) Coding sequences (CDS) are automatically annotated by the Companion tool. (**E**) If CDS surpasses transcript 5′-end, the CDS is manually shortened at the 5′-end. (**F**) After embedding the CDS into the transcript sequence, the gene model emerges with their 5′ and 3′ untranslated regions (UTRs).

**Table 1 genes-14-00866-t001:** The 40 most abundant transcripts in *L. infantum* (JPCM5 strain) promastigotes.

Transcript ID	TPM (SD) *	Encoded Protein
LINF_270018300-T	3059.1 (237.5)	putative histone H1
LINF_060005000-T	3045.2 (381.5)	histone H4
LINF_130010600-T	2872.2 (67.2)	ribosomal protein S12|eS12
LINF_250014900-T	2691.3 (60.7)	cyclophilin A|Cyp19
LINF_190005200-T	2456.7 (220.9)	histone H2B
LINF_350007500-T	2398.6 (210.0)	ribosomal protein L30|eL30
LINF_270006200-T	2379.3 (68.1)	ZFP3
LINF_130009400-T	2362.5 (105.1)	Alba1
LINF_270018800-T	2356.2 (187.1)	putative histone H1
LINF_290024100-T	2262.3 (158.0)	histone H2A
LINF_210028500-T	2160.8 (86.3)	β tubulin
LINF_100016800-T	2086.0 (51.6)	histone H3
LINF_090020900-T	2082.2 (147.6)	histone H2B
LINF_260013700-T	1850.5 (41.8)	ribosomal protein S16|uS9
LINF_360026000-T	1846.1 (131.0)	inosine-guanosine transporter|NT2
LINF_100017000-T	1795.0 (62.1)	histone H3
LINF_260028900-T	1786.9 (52.6)	ribosomal protein L35|uL29
LINF_130017300-T	1772.4 (55.9)	ribosomal protein S4|eS4
LINF_300039100-T	1753.7 (32.4)	ribosomal protein L9|uL6
LINF_200021000-T	1738.7 (61.0)	ribosomal protein S11|uS17
LINF_260013800-T	1708.7 (47.5)	ribosomal protein S16|uS9
LINF_230018500-T	1627.6 (268.5)	protein of unknown function-conserved
LINF_130022200-T	1612.9 (120.3)	ribosomal protein L44|eL42
LINF_260028800-T	1611.7 (37.9)	ribosomal protein L35|uL29
LINF_190005300-T	1588.4 (79.4)	histone H2B
LINF_290023900-T	1584.2 (11.2)	histone H2A
LINF_130017200-T	1582.4 (48.0)	ribosomal protein S4|eS4
LINF_030007300-T	1578.7 (110.0)	ribosomal protein L38|eL38
LINF_130008400-T	1559.2 (65.8)	α tubulin
LINF_350043400-T	1557.6 (56.9)	ribosomal protein L23|uL14
LINF_290024000-T	1557.3 (32.8)	histone H2A
LINF_020005100-T	1554.9 (90.5)	histone H4
LINF_130008800-T	1551.6 (57.8)	α tubulin
LINF_350027400-T	1538.8 (109.4)	kinetoplastid membrane protein-11
LINF_240028300-T	1535.0 (65.4)	ribosomal protein L12|uL11
LINF_350010700-T	1531.8 (49.9)	ribosomal protein L18a|eL20
LINF_230005400-T	1528.6 (150.7)	peroxidoxin mTXNPx
LINF_190005400-T	1523.2 (50.3)	40S ribosomal protein S2|uS5
LINF_280035000-T	1522.6 (156.0)	heat shock protein 70|HSP70-type II
LINF_320009400-T	1520.9 (66.7)	ribosomal protein L17|uL22

* TPM = transcripts per million; SD = standard deviation.

**Table 2 genes-14-00866-t002:** Transcripts with marked different steady-state levels in *L. infantum* and *L. major* promastigotes.

L. infantum Gene	L. major Gene	Ratio ^a^	Gene Product
LINF_350031300	LMJFC_350034400	16.56	hypothetical protein-conserved
LINF_230013000	LMJFC_230013700	16.00	PAP2 superfamily
LINF_360079600	LMJFC_360088700	12.29	FCP1 domain-containing protein
LINF_300027400	LMJFC_300032300	10.85	Zinc finger (C3H1-type) domain-containing protein
LINF_220005200	LMJFC_220005700	9.45	hypothetical protein-conserved
LINF_350026500	LMJFC_350028900	9.41	protein of unknown function-conserved
LINF_010013300	LMJFC_010013800	9.10	potassium channel subunit-like protein
LINF_230009700	LMJFC_230010800	7.06	permease-like protein
LINF_230015000	LMJFC_230016400	6.72	RNA-binding protein|RBP10
LINF_020009800	LMJFC_020010200	6.35	voltage-dependent anion-selective channel
LINF_290021400	LMJFC_290023300	5.41	Nodulin-like
LINF_290020300	LMJFC_290022000	5.19	RNA binding protein
LINF_120010100	LMJFC_120011300	4.97	GF1
LINF_340047300	LMJFC_340050800	4.74	hypothetical protein-conserved
LINF_290019300	LMJFC_290021100	4.59	RNA-binding protein
LINF_260019950	LMJFC_260021700	4.55	protein of unknown function-conserved
LINF_310012900	LMJFC_310013800	4.48	protein of unknown function-conserved
LINF_360076300	LMJFC_360084700	4.45	tartrate-sensitive acid phosphatase
LINF_050016400	LMJFC_050017700	4.36	RNA-binding protein
LINF_350028500	LMJFC_350030900	4.25	protein kinase
LINF_090020400	LMJFC_090020100	0.18	PPPDE peptidase domain-containing protein
LINF_350036100	LMJFC_350040300	0.17	glycerol kinase-glycosomal
LINF_310005000	LMJFC_310005200	0.15	5-methyltetrahydropteroyltriglutamate-homocysteine
			S-methyltransferase
LINF_330008100	LMJFC_330008500	0.15	hypothetical protein-conserved
LINF_120013500	LMJFC_120015000	0.15	surface antigen protein 2
LINF_060018800	LMJFC_060019700	0.14	pteridine transporter
LINF_290007800	LMJFC_290008500	0.14	D-lactate dehydrogenase-like protein
LINF_330038500	LMJFC_330043100	0.14	hypothetical protein-conserved
LINF_020005000	LMJFC_020005200	0.13	Side chain galactosyltransferase (SCG3)
LINF_310008800	LMJFC_310009100	0.05	amino acid transporter|AAT1.3
LINF_220009700	LMJFC_220010200	0.04	hypothetical protein-conserved

^a^ L. infantum TPM/L. major TPM ratio.

## Data Availability

Transcriptomic raw data have been deposited at the European Nucleotide Archive (ENA; http://www.ebi.ac.uk/ena/, accessed on 21 February 2023). The Illumina RNA-seq data, the transcriptome sequences, together with annotations files were uploaded under the Study accession number PRJEB46649 (ERP130857). The Iso-seq raw data and sequences were deposited at ENA under the Study accession number: PRJEB38965.

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
