# Peer review of "Leishmania infantum* (JPCM5) Transcriptome, Gene Models and Resources for an Active Curation of Gene Annotations"

_genes, 2023, doi:10.3390/genes14040866_

Round 1

Reviewer 1 Report

The author has studied Leishmania infantum (JPCM5) transcriptome, gene models and resources for an active curation of gene annotations. I have few concerns given below

1.    In the abstract section of the manuscript, you write lot of common things/introductory part. Kindly add 2-3 sentence of the result part and write 1 line of the aim of the study.

2.    Line no 31, what is Mendeley data; Wikidata?

3.    In the Introduction section add few papers PMID: 29312304 and 30242172

4.    Add a paragraph what you achieved in the study/ novel identification/ scientific society in the last paragraph of the Introduction section.

5.    Add few latest references throughout the manuscript.  

Author Response

-Please, see the attached file

Reviewer 2 Report

This paper primarily describes annotation of gene models in L. infantum using RNAseq. This is a useful data resource. Overall the core description of the resource and its generation appear good, with data availability and methodological information largely clear. However, I'm afraid I find it has some significant failings.

Firstly, it seems to lack recognition that SAS/PAS mapping is something used before in Leishmania - and a key reference is not included (see below). Secondly, a large portion of the results (from ~line 370 onwards) is not very informative and makes big assumptions about correlation of protein abundance with transcript abundance or correlation of transcript abundance across life stages. There is also one section that reads like a targeted attack on the team behind the TriTrypDB database - while I'm sure a personal attack is not the intention, this does need to be changed.

Overall, efforts to improve the quality of the figures, better recognise previous work in this area (and modulate claims of novelty appropriately) and reduce conclusions/analysis drawn from the assumptions mentioned above are needed.

Abstract:

The abstract implies that SAS/PAS mapping has not previously been used as strategy for delineating transcripts in trypanosomatids. This is not correct, having previously been used in Leishmania. A key example methodology paper seems not to have been cited (Fiebig, Gluenz, Carrington and Kelly 2014 Molecular and Biochemical Parasitology) and this has been previously been applied to L. mexicana (Fiebig, Kelly and Gluenz 2015 PLoS Pathogens). It also presents 5'/3' heterogeneity as novel, but that was also shown in Fiebig et al. 2015.

Methods:

Line 124. The methodology for SL sequence identification is likely fine, but comparison and justification vs. using the existing SLaP mapper tool (Fiebig et al. 2014) should be shown. Similarly, for PAS a few lines later.

Versions for various software packages not shown. Exact parameters for some software, eg. SCBN, are not given.

Results:

Line 216. This is a little unclear, is this really known to be a _technical_ issue rather than of biological cause?

Line 244. This has arguably been shown before, with short RNA sequencing reads often spanning PAS and SAS sites even from polyA-selected libraries (eg. Lopez-Escobar et al. 2022). However, I agree that your Iso-Seq data is _far_ stronger than that indirect evidence. It would probably be worthwhile to investigate your short read data for supporting evidence.

Line 295. Conclusions from a single molecule make me concerned.

Line 299. Presumably PacBio reads are long enough to span this region, to provide direct evidence from the genome sequencing too?

Line 306. faint -> fain typo

Line 375. I find this result and Table 1 very trivial, and certainly marginal as to whether it deserves inclusion given that there is no quantitation of protein abundance presented. Eg. Line 428, there is no evidence that the abundant cyclophilin A transcript corresponds to an abundant protein. This applies to all of this paragraph.

Line 462. The data presented does not address this question - there is no plausibility that promastigote transcript abundance will alter disease (caused by the amastigote) nor evidence that promastigote transcript abundance of these genes correlates with amastigote transcript abundance. Much more nuance is needed in the presentation of this data.

Line 539. This comment about the extremely powerful and regularly updated TriTrypDB sounds aggressive, and not supported by my experience working with that team. Please remove this text, it is easily interpreted as a personal attack. With rephrasing, I do agree that there potential value of a wikidata-type repository, however I fail to see how that is any improvement over the comment functionality of TriTrypDB which (again, in my experience) is routinely integrated into the annotations of these genomes.

Line 564. Spurious text.

Figures.

More effort could be put into figure design - eg. Figure 2 is extremely hard to read with the very small font sizes, Figure 3 has strongly squashed text.

Data availability:

Key data availability criteria appear to have been met.

General:

Species names are not consistently italicised. Including the abstract and several places in the methods.

Author Response

-Please, see the attached file
